# Graph-Based Semi-Supervised Learning with Nonignorable Nonresponses

**Fan Zhou[1], Tengfei Li[2], Haibo Zhou[2], Jieping Ye[3], Hongtu Zhu[3,2]**
Shanghai University of Finance and Economics[1], `zhoufan@mail.shufe.edu.cn`
University of North Carolina at Chapel Hill[2], `tengfei_li@med.unc.edu,zhou@bios.unc.edu`
AI Labs, Didi Chuxing [3], `{yejieping,zhuhongtu}@didiglobal.com`

## Abstract

Graph-based semi-supervised learning is very important for many classification tasks, but most existing methods assume that all labelled nodes are randomly sampled. With the presence of nonignorable nonresponse, ignoring all missing nodes can lead to significant estimation bias and handicap the classifiers. To solve this issue, we propose a Graph-based joint model with Nonignorable Missingness (GNM) and develop an imputation and inverse probability weighting estimation approach. We further use graphical neural networks to model nonlinear link functions and then use a gradient descent (GD) algorithm to estimate all the parameters of GNM. We prove the identifiability of the GNM model and validate their predictive performance in both simulations and real data analysis through comparing with models ignoring or misspecifying the missingness mechanism. Our method can achieve up to 7.5% improvement than the baseline model for the document classification task on the Cora dataset.

## 1 Introduction

Graph-based semi-supervised learning problem has been increasingly studied due to more and more real graph datasets. The problem is to predict all the unlabelled nodes in the graph based on only a small subset of nodes being observed. A popular method is to use the graph Laplacian regularization to learn node representations, such as label propagation [25] and manifold regularization [3]. Recently, attention has shifted to the learning of network embeddings [12, 13, 20, 7, 23, 10, 6]. Almost all existing methods assume that the labelled nodes are randomly selected. However, the probability of missingness may depend on the unobserved data after conditioning on the observed data in the real world. For example, when predicting the traffic volume of a road network, sensors used to collect data are usually set up at intersections with large traffic flow. A researcher is more likely to label the documents in a citation network that fall into the categories which he or she is more familiar with. In these cases, non-responses may be missing not at random (MNAR). Ignoring nonignorable nonresponses may be unable to capture the representativeness of remaining samples, leading to significant estimation bias.

Modeling non-ignorable missingness is challenging because the MNAR mechanism is usually unknown and may require additional model identifiability assumptions [5, 14, 21]. A popular method assigns the inverse of estimated response probabilities as weights to the observed nodes [16, 4], but these procedures are designed for the missing at random (MAR) mechanism instead of MNAR. Another method is to impute missing data by using observed data [18, 19, 11]. Some more advanced methods [24, 21] have been proposed to estimate the non-ignorable missingness using external data [8], but such data is often unavailable in many applications, making these methods infeasible. Moreover, all these methods are built on simple regressions and are not directly applied to graphs.

In this paper, we develop a Graph-based joint model with Nonignorable Missingness (GNM) by assigning inverse response probability to labelled nodes when estimating the target classifier or regression. To model the non-ignorable missingness, we propose a deep learning based exponential tilting model to utilize the strengths of neural networks in function approximation and representation learning. The main contributions of this paper can be summarized as follows:

- To the best of our knowledge, we are the first to consider the graph-based semi-supervised learning problem in the presence of non-ignorable nonresponse and try to solve the problem from the perspective of missing data.

- We introduce a novel identifiability in prediction-equivalence quotient (PEQ) space for neural network architectures and its easily checked sufficient conditions.

- Different from traditional statistical methods which extract features and fit the prediction model separately, we propose a novel joint estimation approach by integrating the inverse weighting framework with a modified loss function based on the imputation of non-response. It is easy to implement in practice and is robust to the normality assumption when the node response is continuous.

- We use gradient descent (GD) algorithm to learn all the parameters, which works for traditional regression model as well as for modern deep graphical neural networks.

- We examine the finite sample performance of our methods by using both simulation and real data experiments, demonstrating the necessity of 'de-biasing' in acquiring unbiased prediction results on the testing data under the non-ignorable nonresponse setting.

## 2 Model Description

Let $G = (V, E, A)$ be a weighted graph, where $V = \{v_1, \ldots, v_N\}$ denotes the vertex set of size $|V| = N$, $E$ contains all the edges, and $A$ is an $N \times N$ adjacency matrix. The $N$ vertexes make up the whole population with only a small subset of vertexes being labelled. We introduce some important notations as follows:

(i). $\mathbf{x} = [x_1, x_2, \ldots, x_N]^T \in R^{N \times p}$ is a fully observed input feature matrix of size $N \times p$ with each $x_i \in R^p$ being a $p \times 1$ feature vector at vertex $v_i$.

(ii). $Y = (y_1, y_2, \ldots, y_N)^T$ is a vector of vertex responses, which is partially observed subject to missingness, and $y_i$ can be either categorical or continuous.

(iii). $A \in R^{N \times N}$ is the adjacency matrix (binary or weighted), which encodes node similarity and network connectivity. Specifically, $a_{ij}$ represents the edge weight between vertexes $v_i$ and $v_j$.

(iv). $r_i \in \{0, 1\}$ is a "labeling indicator", that is $y_i$ is observed if and only if $r_i = 1$. Let $R = \{1, \ldots, n\}$ denote the set of labelled vertexes and $R^c = \{n + 1, \ldots, N\}$ defines the subsample of non-respondents for which the vertex label is missing.

(v) $\mathscr{G}^A(\mathbf{x}; \theta_g) \in R^{N \times q}$ denotes a $q \times 1$ vector of unknown function of $\mathbf{x}$, which can be a deep neural network incorporating the network connectivity $A$, such as a multi-layer GCN [10] or GAT [22].

In this paper, we consider an non-ignorable response mechanism, where the indicator variable $r_i$ depends on $y_i$ (which is unobserved when $r_i = 0$). It is assumed that $r_i$ follows a Bernoulli distribution as follows:
$$r_i | (y_i, h(x_i; \theta_h)) \sim \text{Bernoulli}(\pi_i), \tag{1}$$
where $h(x_i; \theta_h)$ is an unknown parametric function of $x_i$ and $\pi(y_i, h(x_i; \theta_h)) = P(r_i = 1 | y_i, h(x_i; \theta_h))$ is the probability of missingness for $y_i$. Given $\mathscr{G}^A(\mathbf{x}; \theta_g)$, $y_i$ and $y_j$ are assumed to be independent and given $y_i$ and $h(x_i; \theta_h)$, $r_i$ and $r_j$ are assumed to be independent for $i \neq j$. Furthermore, an exponential tilting model is proposed for $\pi_i$ as follows:

$$\pi(y_i, h(x_i; \theta_h)) = \pi(y_i, h(x_i; \theta_h); \alpha_r, \gamma, \phi) = \frac{\exp\{\alpha_r + \gamma^T h(x_i; \theta_h) + \phi y_i\}}{1 + \exp\{\alpha_r + \gamma^T h(x_i; \theta_h) + \phi y_i\}}. \tag{2}$$

Our question of interest is to unbiasedly learn an outcome model $Y | \mathbf{x}$. Without loss of generality, when $y$ is continuous, we consider a linear model given by

$$Y = \alpha + \mathscr{G}^A(\mathbf{x}; \theta_g)\beta + \epsilon, \tag{3}$$

where $\epsilon = (\epsilon_1, \cdots, \epsilon_N)^T \sim N(\mathbf{0}, \sigma^2 I)$ and $\epsilon \perp \mathbf{x}$ is the error term with zero unconditional mean, that is, $E(\epsilon_i) = 0$. In this case, dropping out missing data can lead to strongly biased estimates when $r$ depends on $y$. The parameter estimates will not be consistent since $E\{\epsilon_i | r_i = 1\}$ and $E\{\epsilon_i \mathscr{G}^A(\mathbf{x}; \theta_g)_i | r_i = 1\}$ are not zero. The missing values could not be imputed even if we would have consistent estimates since

$$E\{y_i | r_i = 0, \mathscr{G}^A(\mathbf{x}; \theta_g)_i; \alpha, \beta\} = \frac{E\{y_i(1 - r_i)|\mathscr{G}^A(\mathbf{x}; \theta_g)_i; \alpha, \beta\}}{1 - P(r_i = 1|\mathscr{G}^A(\mathbf{x}; \theta_g)_i; \alpha, \beta)} \quad (4)$$

$$= \alpha + \beta^T \mathscr{G}^A(\mathbf{x}; \theta_g)_i - \frac{\text{cov}(y_i, \pi_i | \mathscr{G}^A(\mathbf{x}; \theta_g)_i; \alpha, \beta)}{1 - E(\pi_i | \mathscr{G}^A(\mathbf{x}; \theta_g)_i; \alpha, \beta)} \neq \alpha + \beta^T \mathscr{G}^A(\mathbf{x}; \theta_g)_i.$$

When $y$ is a $K$-class discrete variable, we consider an multicategorical logit model as follow:

$$P(y_i = k|\mathscr{G}^A(\mathbf{x}; \theta_g)_i; \alpha_k, \beta_k) = \exp(\alpha_k + \beta_k^T \mathscr{G}^A(\mathbf{x}; \theta_g)_i) / \sum_{j=1}^{K} \exp(\alpha_j + \beta_j^T \mathscr{G}^A(\mathbf{x}; \theta_g)_i) \quad \forall k \quad (5)$$

Therefore, we can define a joint model of (1) and (3) (or (1) and (5)), called Graph-based joint model with Nonignorable Missingness (GNM) to obtain the unbiased estimation of $Y|\mathbf{x}$.

## 3 Estimation

We examine several important properties, such as identifiability, of GNM and its estimation algorithm in this section.

### 3.1 Identifiability

We consider the identifiability property of GNM. Let $Y = (y_{obs}^T, y_{mis}^T)^T$ and $J = (R, R^c)$. The joint probability density function (pdf) of the observed data is given by

$$f(y_{obs}, J|\mathbf{x}) = f(y_1, y_2 \ldots, y_n, r_1, \ldots, r_N|\mathbf{x}) = \prod_{i=1}^{n} f(y_i, r_i|\mathbf{x}) \prod_{i=n+1}^{N} \int f(y_i, r_i|\mathbf{x}) dy_i. \quad (6)$$

Based on the assumptions of $r_i|(y_i, h(x_i))$ and $y_i|\mathscr{G}^A(\mathbf{x}; \theta_g)_i$, (6) is equivalent to

$$\prod_i [P(r_i = 1|y_i, h(x_i; \theta_h)) f(y_i|\mathscr{G}^A(\mathbf{x}; \theta_g)_i)]^{r_i} [1 - \int P(r_i = 1|y, h(x_i; \theta_h)) f(y|\mathscr{G}^A(\mathbf{x}; \theta_g)_i) dy]^{1-r_i}.$$

$$(7)$$

The GNM model is called identifiable if for different sets of parameters $(\theta_h, \theta_g)$, $P(r_i = 1|y_i, h(x_i; \theta_h)) f(y_i|\mathscr{G}^A(\mathbf{x}; \theta_g)_i)$ are different functions of $(y_i, \mathbf{x})$. The identifiability implies that in a positive probability, the global maximum of (7) is unique.

However, identifiability may fail for many neural network models. For example, the identifiability of parameters in (1) is one of the necessary conditions for model identifiability, which can fail for the Relu network. Specifically, we have

$$\text{Logit}[P(r_i = 1|y_i, h(z_i; \beta_r)); \gamma] = \alpha_r + \gamma \text{Relu}(z_i \beta_r) + \phi y_i = \text{Logit}[P(r_i = 1|y_i, h(z_i; 2\beta_r)); \gamma/2].$$

Fortunately, this type of non-identifiability does not create any prediction discrepancy, since under GNM, the prediction of $y$ given $x$ is exactly the same for different $(\gamma, \theta_h, \beta, \theta_g)$ and $(\gamma', \theta_h', \beta', \theta_g')$ if we have

$$\gamma^T h(x; \theta_h) = \gamma'^T h(x; \theta_h'), \text{ and } \mathscr{G}^A(\mathbf{x}; \theta_g)\beta = \mathscr{G}^A(\mathbf{x}; \theta_g')\beta'. \quad (8)$$

In consideration of the prediction equivalence, a more useful definition of identifiability is given in the following. Let $f(y_i|\mathscr{G}^A(\mathbf{x})_i; \theta_y) = f(y_i|\mathscr{G}^A(\mathbf{x}; \theta_g)_i; \alpha, \beta)$ and $P(r_i = 1|y_i, h(z_i); \theta_r) = P(r_i = 1|y_i, h(z_i; \theta_h); \alpha_r, \gamma, \phi)$, where $\theta_y = (\alpha, \beta, \theta_g)$ and $\theta_r = (\alpha_r, \gamma, \phi, \theta_h)$ contain unknown parameters in the outcome model $Y|\mathbf{x}$ and the missing data model $r|(y, z)$. The $\mathcal{D}(\theta_y) \otimes \mathcal{D}(\theta_r)$ denotes the domain of $(\theta_y, \theta_r)$, where $\otimes$ is the tensor product of two spaces.

**Definition 3.1.** *Under GNM, we call $(\theta_y, \theta_r)$ is **equivalent** to $(\theta'_y, \theta'_r)$, denoted by*

$$(\theta_y, \theta_r) \sim (\theta'_y, \theta'_r),$$

*if (8) holds and $\alpha' = \alpha, \alpha'_r = \alpha_r$ and $\phi' = \phi$, where $\theta_y = (\alpha, \beta, \theta_g), \theta_r = (\alpha_r, \gamma, \phi, \theta_h), \theta'_y = (\alpha', \beta', \theta'_g)$, and $\theta'_r = (\alpha'_r, \gamma', \phi', \theta'_h)$. The equivalence class of an element $(\theta_y, \theta_r)$ is denoted by $[\![(\theta_y, \theta_r)]\!]$, defined as the set*

$$[\![(\theta_y, \theta_r)]\!] = \{(\theta'_y, \theta'_r) \in \mathcal{D}(\theta_y) \otimes \mathcal{D}(\theta_r) | (\theta'_y, \theta'_r) \sim (\theta_y, \theta_r)\},$$

*and the set of all equivalent classes is called the **Prediction-Equivalent Quotient** (PEQ) space, denoted by $S = \mathcal{D}(\theta_y) \otimes \mathcal{D}(\theta_r)/\sim$. The GNM model is called identifiable on the PEQ space iff that*

$$f(y|\mathscr{G}^A(\boldsymbol{x})_i; \theta_y)P(r = 1|y, h(x_i); \theta_r) = f(y|\mathscr{G}^A(\boldsymbol{x})_i; \theta'_y)P(r = 1|y, h(x_i); \theta'_r)$$

*holds for all $\boldsymbol{x}, y$ implies $(\theta_y, \theta_r) \sim (\theta'_y, \theta'_r)$.*

Different from identifiability on the parameter space, the identifiability on the PEQ space implies the uniqueness of the prediction given **x** instead of parameter estimation. It is applicable to complex architecture that focuses more on prediction than parameter. The following is an example which is not identifiabile on both parameter space and PEQ space.

**Example 1**. Let $\mathscr{G}^A(x; \theta_g) = x, h(x; \theta_h) = x, y_i \sim N(\mu + x\beta, 1)$, and $P(r_i = 1|y_i) = [1 + \exp(-\alpha_r - x\gamma - \phi y_i)]^{-1}$ with unknown real-valued $\alpha_r, \gamma, \phi, \mu$ and $\beta$, and thus

$$P(r_i = 1|y_i, h(x_i))f(y_i|\mathscr{G}^A(\mathbf{x})_i) = \frac{\exp[-(y_i - \mu - x_i\beta)^2/2]}{\sqrt{2\pi}[1 + \exp(-\alpha_r - \phi y_i - \gamma x)]}. \tag{9}$$

In this case, two different sets of parameters $(\alpha_r, \gamma, \phi, \mu, \beta)$ and $(\alpha'_r, \gamma', \phi', \mu', \beta')$ produce equal (9) values if $\alpha_r = -(\mu^2 - \mu'^2)/2, \beta' = \beta, \phi = \mu' - \mu, \gamma = \beta(\mu - \mu'), \alpha'_r = -\alpha_r, \phi' = -\phi$, and $\gamma = -\gamma'$. The observed likelihood is only identifiable with ignorable missingness, i.e. $\phi = \phi' = 0$.

Additional conditions are required to ensure the identifiability of GNM on the PEQ space.

**Theorem 3.1.** *Assume three conditions as follows.*

*(A1) For all $\theta_g$, there exist $(\boldsymbol{x}_1, \boldsymbol{x}_2)$ such that $\mathscr{G}^A(\boldsymbol{x}_1; \theta_g)_i \neq \mathscr{G}^A(\boldsymbol{x}_2; \theta_g)_i$ for each i; $\beta \neq 0$ holds.*

*(A2) For all $\theta_g$ and z, there exists $(\boldsymbol{u}_1, \boldsymbol{u}_2)$ such that $\mathscr{G}^A([\boldsymbol{z}, \boldsymbol{u}_1]; \theta_g)_i \neq \mathscr{G}^A([\boldsymbol{z}, \boldsymbol{u}_2]; \theta_g)_i$ for each i; and $\beta \neq 0$ holds.*

*(A3) For all $\theta_h$, there exists $(z_1, z_2)$ such that $h(z_1; \theta_h) \neq h(z_2; \theta_h)$; and $\gamma \neq 0$ holds.*

*The GNM model (1) and (5) is identifiable on the PEQ space under Condition (A1). Suppose that there exists an instrumental variable **u** in $\boldsymbol{x} = [\boldsymbol{z}, \boldsymbol{u}]$ such that $f(y_i|\mathscr{G}^A(\boldsymbol{x})_i)$ depends on **u**, whereas $P(r_i = 1|y_i, h(x_i))$ does not. Then the GNM model (1) and (3) is identifiable on the PEQ space under Conditions (A2) and (A3).*

Regularity conditions (A1)∼(A3) are easy to satisfy.

## 3.2 Estimation Approach

It is not easy to directly maximize the full likelihood function (6) in practice since it can be extremely difficult to compute its integration term. On the other hand, the normality assumption of the error term can be restrictive for GNM consisting of (1) and (3). Therefore, we propose a doubly robust (DR) estimation approach to alternatively obtain the Inverse Probability Weighted Estimator (IPWE) of $\theta_y$ and imputation estimator of $\theta_r$ [15, 1].

**Inverse Probability Weighted Estimator (IPWE) of $\theta_y$**

With $\pi(y_i, h(x_i); \theta_r)$ estimated by $\pi(y_i, h(x_i); \widehat{\theta}_r)$, the Inverse Probability Weighted Estimator (IPWE) of $\theta_y$ can be obtained by minimizing the weighted cross-entropy loss

$$\mathscr{L}_1(\theta_y|\widehat{\theta}_r) = -\sum_i \frac{r_i}{\pi(y_i, h(x_i); \widehat{\theta}_r)} \sum_{k=1}^{K} 1(y_i = k)\log(P(y_i = k|\mathscr{G}^A(\mathbf{x})_i; \theta_y)) \tag{10}$$

when $Y|\mathbf{x}$ follows (5) or by minimizing the weighted mean squared error (MSE)

$$\mathscr{L}_1(\theta_y|\widehat{\theta}_r) = \sum_i \frac{r_i}{\pi(y_i, h(x_i); \widehat{\theta}_r)} \{y_i - \alpha - \beta^T \mathscr{G}^A(\mathbf{x}; \theta_g)_i)\}^2 \qquad (11)$$

when $Y$ is continuous. The estimation equation (11) is robust with respect to the normality assumption. If $\pi(y_i, h(x_i); \theta_r)$ is correctly specified, the IPW estimator of $\theta_y$ that solves $\partial \mathscr{L}_1(\theta_y|\widehat{\theta}_r)/\partial \theta_y = 0$ is consistent and converges to $\theta_y$ according to the following theorem.

**Theorem 3.2.** *If $\theta_r$ is known, then a given estimating function $l(y_i, \mathscr{G}^A(\mathbf{x})_i; \theta_y)$ with $E_{\theta_y}\{\sum_i l(y_i, \mathscr{G}^A(\mathbf{x})_i; \theta_y)\} = 0$ satisfies*

$$E_{\theta_y}\{\sum_i \frac{r_i}{\pi(y_i, h(x_i); \theta_r)} l(y_i, \mathscr{G}^A(\mathbf{x})_i; \theta_y)\} = 0.$$

**Imputation estimator of $\theta_r$**

With the estimated $f(Y|\mathscr{G}^A(\mathbf{x}; \widehat{\theta}_g))$, we could obtain an estimator of $\theta_r$ by minimizing

$$\mathscr{L}_2(\theta_r|\widehat{\theta}_y) = -\sum_{r_i=1} \log(\pi(y_i, h(x_i); \theta_r)) - \sum_{r_i=0} \log(1 - E\{\pi(y_i, h(x_i); \theta_r)|\mathbf{x}; \widehat{\theta}_y\}), \qquad (12)$$

where $\pi(y_i, h(x_i); \theta_r) = P(r_i = 1|y_i, h(x_i); \theta_r)$ and $E\{\pi(y_i, h(x_i))|\mathbf{x}; \widehat{\theta}_y\} = \int P(r_i = 1|y, h(x_i); \theta_r) f(y|\mathscr{G}^A(\mathbf{x})_i; \widehat{\theta}_y) dy$. One advantage of our proposed joint estimation approach is that $E(\pi(y_i, h(x_i); \theta_r)|\mathbf{x})$ can be easily approximated by the empirical average of a set of random draws at the nodes with missing $y$ as the imputed responses:

$$E\{\pi(y_i, h(x_i); \theta_r)|\mathbf{x}; \theta_y\} = \int P(r_i = 1|y, h(x_i); \theta_r) f(y|\mathscr{G}^A(\mathbf{x})_i; \theta_y) dy \approx B^{-1} \sum_b \pi(y_{ib}, h(x_i); \theta_r),$$

where $\{y_{ib}\}_{b=1}^B \overset{iid}{\sim} f(y|\mathscr{G}^A(\mathbf{x})_i; \widehat{\theta}_y)$. Thus, we can get an unbiased estimate of (12) by replacing the expectation by an empirical mean over samples generated from $f(y|\mathscr{G}^A(\mathbf{x})_i; \widehat{\theta}_y)$ as follows:

$$\widetilde{\mathscr{L}}_2(\theta_r|\widehat{\theta}_y) = -\sum_{r_i=1} \ln(\pi(y_i, h(x_i); \theta_r)) - \sum_{r_i=0} \log(1 - B^{-1} \sum_{y_{ib} \sim f(y|\mathscr{G}^A(\mathbf{x})_i; \widehat{\theta}_y)} \pi(y_{ib}, h(x_i); \theta_r)),$$

$$(13)$$

the gradient of which can be expressed as

$$\nabla_{\theta_r} \widetilde{\mathscr{L}}_2(\theta_r|\widehat{\theta}_y) = -\sum_{r_i=1} \frac{\nabla_{\theta_r} \pi_i}{\pi_i} + \sum_{r_i=0} \frac{B^{-1} \sum_b \nabla_{\theta_r} \pi(y_{ib}, h(x_i); \theta_r)}{1 - B^{-1} \sum_b \pi(y_{ib}, h(x_i); \theta_r)}. \qquad (14)$$

The imputation estimator of $\theta_r$ by minimizing $\mathscr{L}_2(\theta_r|\theta_y)$ is consistent when $f(Y|\mathscr{G}^A(\mathbf{x}; \theta_g))$ is correctly specified. The overall estimation procedure is schematically depicted in Figure 1.

### 3.3 Algorithm

In this subsection, we provide more details of our proposed imputation and IPW estimation approach about how to jointly estimate $\theta_y$ and $\theta_r$ by alternatively minimizing the conditional loss functions $\mathscr{L}_1(\theta_y|\widehat{\theta}_r)$ and $\widetilde{\mathscr{L}}_2(\theta_r|\widehat{\theta}_y)$ in practice. Specifically, we update $\theta_y$ and then $\theta_r$ with $\theta_y^{(e+1)} = \arg\min_{\theta_y} \mathscr{L}_1(\theta_y|\theta_r^{(e)})$ and $\theta_r^{(e+1)} = \arg\min_{\theta_r} \widetilde{\mathscr{L}}_2(\theta_r|\theta_y^{(e+1)})$ in order at each epoch, where $\theta_r^{(e)}$ and $\theta_y^{(e+1)}$ are the estimates of $\theta_r$ and $\theta_y$ obtained at the $e$-th and $(e+1)$-th epoch, respectively. We use the gradient descent (GD) algorithm to learn all the parameters in $\theta_r$ and $\theta_y$, while incorporating the network architecture of $\mathscr{G}^A(\mathbf{x}; \theta_g)$ and $h(x; \theta_h)$.

Without specifying the normal assumption when $y_i$ is continuous, we replace the random draw $y_{ib}^{(e)}$ in (13) by the expectation of $\beta_0 + \beta_1^T \mathscr{G}^A(\mathbf{x}; \theta_g^{(e)})_i$ at the $e$-th epoch. It can be seen as an approximation obtained by linearizing $\pi(y_i, h(x_i))$ using a Taylor series expansion and taking the expectation of the first two terms [2]:

$$E\{\pi(y_i, h(x_i))|\mathbf{x}; \theta_y^{(e)}\} \approx \pi(E(y_i|\mathbf{x}; \theta_y^{(e)}), h(x_i)) = \pi(\beta_0 + \beta_1^T \mathscr{G}^A(\mathbf{x}; \theta_g^{(e)})_i, h(x_i)).$$

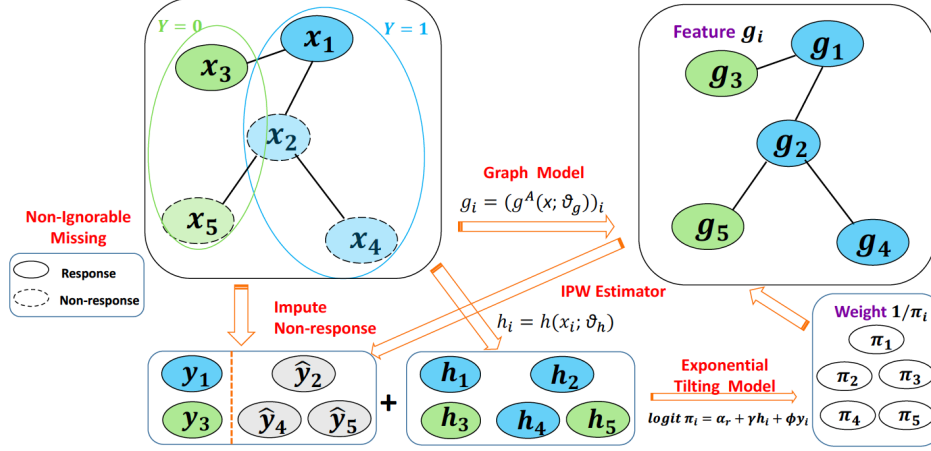

Figure 1: General Picture of the Joint Estimation Approach

In this case, it is equivalent to let $B = 1$ and the sample size, i.e. the total number of nodes will be fixed at each training epoch. Based on simulations and real experiments below, this simplification still outperforms the baseline models with a significant improvement in the prediction accuracy on non-response nodes.

The details of the algorithm are described in five steps as follows:

1. Determine the initial value of the response probability $\pi_i^{(0)}$ (or $\theta_r^{(0)}$). For example, we can let $\pi_i^{(0)} = 1$ for all the labelled vertexes ($r_i = 1$).

2. Let $e = 1$, where $e$ represents the number of epoch. We update $\theta_y$ based on $\pi_i^{(0)}$ obtained from the previous epoch by minimizing the loss function in (10) using GD. At the $i$-th iteration within the $e$-th epoch, we update $\theta_y$ as follows:
$$\theta_y^{(e,i+1)} \leftarrow \theta_y^{(e,i)} - \gamma_0 \nabla_{\theta_y} \mathscr{L}_1(\theta_y | \theta_r^{(e-1)}), \tag{15}$$
where $\gamma_0$ is the learning rate and $\mathscr{L}_1(\theta_y | \theta_r^{(e-1)})$ represents the loss function based on $\pi_i^{(e-1)} = \pi_i(y_i, h(x_i); \theta_r^{(e-1)})$. We denote the updated $\theta_y$ as $\theta_y^{(e)}$ after $M^{(e)}$ iterations.

3. Impute $y_i$ for all the unlabelled nodes $r_i = 0$ using $y_i^{(e)} = \beta_0^{(e)} + \mathscr{G}^A(\mathbf{x}; \theta_g^{(e)})_i^T \beta_1^{(e)}$ for the continuous case and sampling $y_i^{(e)}$ from distribution $P(y_i | \mathscr{G}^A(\mathbf{x})_i; \theta_y^{(e)})$ otherwise.

4. We use GD to update $\theta_r$. Specifically, at the $j$-th iteration, we have
$$\theta_r^{(e,j+1)} \leftarrow \theta_r^{(e,j)} - \gamma_1 \nabla_{\theta_r} \widetilde{\mathscr{L}_2}(\theta_r | \theta_y^{(e)}) \tag{16}$$
with the initial start $\theta_r^{(e,0)}$ equal to $\theta_r^{(e-1)}$, and $\gamma_1$ is the learning rate. After convergence, we can get the estimate of $\theta_r$ denoted as $\theta_r^{(e)}$ at the end of this training epoch. Then we update the sampling weight $\pi_i^{(e)}$ based on $P(r_i = 1 | y_i, h(x_i); \theta_r^{(e)})$ for all labelled vertexes.

5. Stop once convergence has been achieved, otherwise let $e = e + 1$ and return to step 3.

The convergence criterion is that whether the imputed unlabelled vertexes at epoch $e$ only slightly differ from those at epoch $(e - 1)$. In other words, the iteration procedure is stopped if
$$\sum_{r_i=0} |y_i^{(e)} - y_i^{(e-1)}| / \sum_i \mathbf{1}(r_i = 0) \leq \varepsilon$$
We let $M_0$ and $M_1$ be the maximal number of allowed internal iterations at each epoch for updating $\theta_y$ and $\theta_r$, respectively. For more details, you can refer to the Algorithm 1 in the supplements.

Theoretically, the complexity (for example one layer GCN) in Steps 2 and 3 is $O(|E|pq)$ at each epoch according to [10], where $|E| < N^2$ is the number of edges. Moreover, the complexity of Step 4 is $O(Np)$ when $h$ is a one fully connected layer.

## 4 Experiments

In this section, simulations and one real data analysis are conducted to evaluate the empirical performance of our proposed methods and a baseline model, which ignores the non-response (SM). Our GNM model reduces to SM when it only contains the outcome model $Y|x$ given in (3) (or (5)) with the weights in loss (10) being 1 for all samples. In the real data part, GNM is also compared with the model with a misspecified ignorable missing mechanism, and some other state-of-art 'de-biasing' methods. In the simulation part, we simulate the node response $y$ based on (3) and generate the labelled set by the exponential tilting model (1). For the real data analysis, we evaluate all the compared models by a semi-supervised document classification on the citation network-Cora with non-ignorable non-response.

In this paper, we use GCN to learn the latent node representations $\mathscr{G}^A(\mathbf{x})$ with the layer-wise propagation defined as

$$H^{(l+1)} = f(H^{(l)}, A) = \sigma(\widehat{D}^{-\frac{1}{2}} \widehat{A} \widehat{D}^{-\frac{1}{2}} H^{(l)} W^{(l)}), \tag{17}$$

where $\widehat{A} = A + I$, in which $I$ is an identity matrix, and $\widehat{D}$ is the diagonal vertex degree matrix of $\widehat{A}$. The $W^{(l)}$ is a weight matrix for the $l$-th layer and $\sigma(\cdot)$ is an non-linear activation function. $H^{(0)} = \mathbf{x}$ is the initial input and $\mathscr{G}^A(\mathbf{x}) = H^{(2)} \in R^{N \times \bar{p}}$ is the output of the second layer-wise propagation. To be fair, we let $\mathscr{G}^A(\mathbf{x})$ be a 2-layer GCN model for all compared approaches.

### 4.1 Simulations

We consider a network data generated by $|V| = 2708$ vertexes together with a binary adjacency matrix $A$. $\mathbf{x} \in R^{2708 \times 1433}$ denotes the fully observed input features which is a large-scale sparse matrix. Both $A$ and $\mathbf{x}$ are obtained from the Cora dataset. The node response is simulated from the following model:

$$y_i = \beta_0 + \beta_1^T \mathscr{G}^A(\mathbf{x})_i + \epsilon_i, \tag{18}$$

where $\epsilon_i \sim N(0, \sigma^2)$ and $\mathscr{G}^A(\mathbf{x})$ is the output of a 2-layer GCN model. We let response probability $\pi$ depend on the unobserved vertex response $y$ only , and (1) is simplified to

$$\pi_i \equiv P(r_i = 1|y_i) = \frac{\exp\{\alpha_r + \phi y_i\}}{1 + \exp\{\alpha_r + \phi y_i\}}. \tag{19}$$

In this case, the instrumental variable $\mathbf{u}$ is exactly $\mathbf{x}$ itself, and the identifiability automatically holds according to Theorem 3.1. All $\beta$'s in (18) are sampled from uniform distribution $U(0, 1)$. The $\alpha_r$ and $\phi$ were selected to make the overall missing proportion be approximately 90%. The labelled subset are randomly split into training and validation sets, while the remaining non-response nodes build the testing set. We train all the compared models for a maximum of 200 epochs ($E = 200$) using Adam [9] with a learning rate 0.05 and make predictions $\widehat{y}_i$ for each testing vertex. Training is stopped when validation loss does not decrease in 15 consecutive iterations. We keep all other model settings used by [10] and fix the unit size of the first hidden layer to be 16.

Table 1 summarizes the estimation results under different $(\bar{p}, \sigma)$ combinations, where root mean squared error (RMSE) and Mean absolute percentage error (MAPE) are computed between the true node response $y$ and prediction $\widehat{y}$ over the 50 runs. We can clearly see that GNM outperforms SM under all the four settings with much smaller mean RMSEs and MAPEs. Moreover, GNM is more stable than SM with smaller estimation variance.

### 4.2 Real Data Analysis

For the real data analysis, we modify the Cora to a binary-class data by merging the six non-'Neural Network' classes together. The global prevalence of two new classes are $(0.698, 0.302)$ with $N_0 = \#\{y = 0\} = 1890$ and $N_1 = \#\{y = 1\} = 818$, respectively.

Two missing mechanisms are considered. A simple setup is the same as (19). In this case, we compare our method with the inverse weighting approach proposed by [17]. We let the two functions of $x$ required to estimate $\pi$ under their framework to be the constant 1 and the first principle component (PC) score, which is more stable compared to other functions such as a general $x_j$ or $\sum_j x_j$. In a

| $\bar{p}$ | $\sigma$ | Method | Metric | Mean | SD |
|---|---|---|---|---|---|
| 4 | 0.5 | SM | RMSE | 1.1925 | 6.43e-1 |
| | | | MAPE | 0.2932 | 2.01e-1 |
| | | GNM | RMSE | 0.6983 | 1.28e-2 |
| | | | MAPE | 0.1995 | 1.00e-2 |
| | 1 | SM | RMSE | 1.6185 | 8.58e-2 |
| | | | MAPE | 0.3104 | 4.73e-2 |
| | | GNM | RMSE | 1.2103 | 4.81e-2 |
| | | | MAPE | 0.2263 | 2.28e-2 |
| 16 | 0.5 | SM | RMSE | 0.7923 | 9.94e-2 |
| | | | MAPE | 0.2014 | 2.42e-2 |
| | | GNM | RMSE | 0.6015 | 2.17e-2 |
| | | | MAPE | 0.1672 | 1.90e-2 |
| | 1 | SM | RMSE | 1.4212 | 2.14e-1 |
| | | | MAPE | 0.2129 | 1.05e-2 |
| | | GNM | RMSE | 1.1316 | 6.04e-2 |
| | | | MAPE | 0.1849 | 4.62e-3 |

Table 1: Mean RMSEs and MAPEs by GNM and SM based on simulated data sets

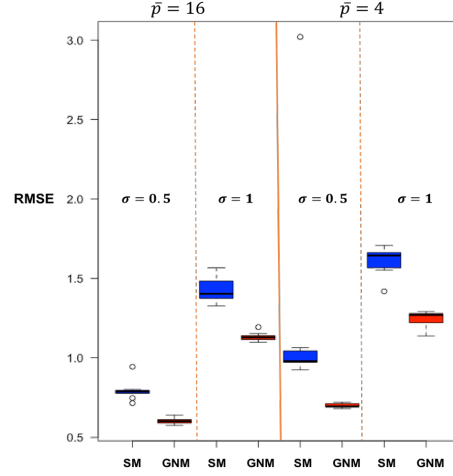

Figure 2: Boxplot of RMSEs in real data analysis

more complicated setup, the labelled nodes are generated based on

$$\pi_i \equiv P(r_i = 1 | y_i, h(x_i)) = \frac{\exp\{\alpha_r + \gamma^T h(x_i) + \phi y_i\}}{1 + \exp\{\alpha_r + \gamma^T h(x_i) + \phi y_i\}}, \tag{20}$$

where $h(x_i) = \exp(\sum_j x_{ij}/a_0 - a_1) - (\sum_j x_{ij} - a_2)/a_3$ with value range being $[0, 1]$. The explicit form of $h(x)$ is assumed to be unknown and we use a multi-layer perceptron to approximate it. The network has two hidden layers with 128 and 64 units. respectively, and we use the 'tanh' activation for the final output layer. As a comparison, we also include the results when the 'non-ignorable' missingness is over-simplified to the 'ignorable' one (GIM). We let $n_k = \#\{(y_i = k) \wedge (r_i = 1)\}$, and use $\lambda$ to denotes the size ratio between the two groups of labelled nodes, i.e. $n_1/n_0$. We carry out more experiments on other datasets including 'Citeseer', and explore the finite sample performance of our method using other state-of-art architecture such as GAT [22]. More details are provided in the supplementary materials. [1]

| | | Accuracy | |
|---|---|---|---|
| $\lambda$ | Method | Mean | SD |
| 1 | SM | 0.8683 | 1.98e-2 |
| | Rosset | 0.8514 | 5.19e-2 |
| | **GNM** | **0.8947** | **6.47e-3** |
| 1.5 | SM | 0.8458 | 2.21e-2 |
| | Rosset | 0.8311 | 7.09e-2 |
| | **GNM** | **0.8908** | **1.26e-2** |
| 2 | SM | 0.8052 | 3.26e-2 |
| | Rosset | 0.8193 | 6.05e-2 |
| | **GNM** | **0.8648** | **2.54e-2** |

Table 2: Mean Prediction Accuracy for the simple setup by each method

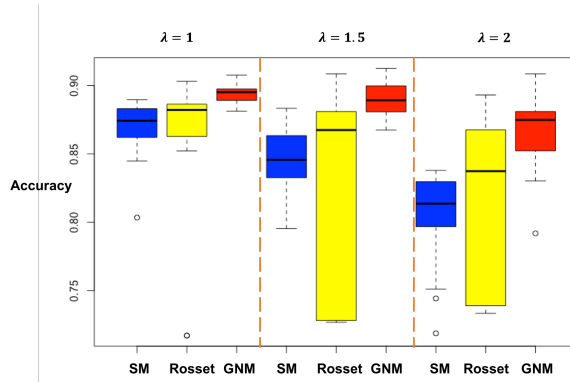

Figure 3: Boxplot Prediction Accuracy for the simple setup

Results are summarized in Tables 2 and 3. Reported values represent the average classification accuracy on testing data by 50 replications with re-sampling allowed. In each setup, two 'de-biasing' methods including our approach are compared with SM. We adjust $\alpha$ and $\beta$ to make the size of training set be around 120 for each sub-setting. Increasing $\lambda$ reduces the number of included $y = 0$ nodes in the training set, leading to an insufficient learning power and thus a lower overall classification

|  | | Accuracy | |
|---|---|---|---|
| $\lambda$ | Method | Mean | SD |
| 1 | SM | 0.8663 | 1.21e-2 |
|  | GIM | 0.8713 | 1.52e-2 |
|  | **GNM** | **0.8961** | **1.18e-2** |
| 2 | SM | 0.8141 | 2.34e-2 |
|  | GIM | 0.8291 | 2.79e-2 |
|  | **GNM** | **0.8669** | **1.63e-2** |

Table 3: Mean Prediction Accuracy for the complicated setup by each method

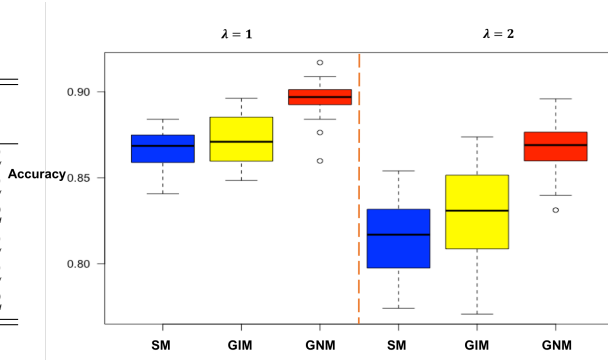

Figure 4: Boxplot of Prediction Accuracy for the complicated setup

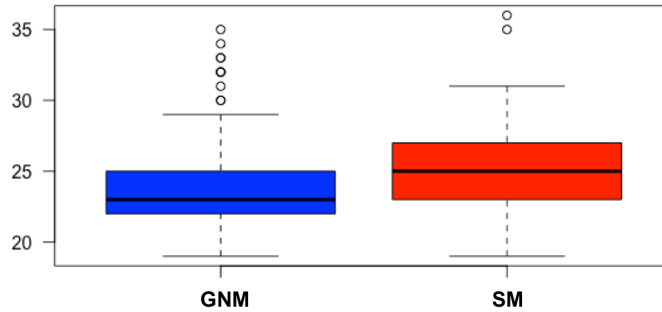

Figure 5: Number of iteration times for GNM and SM at each epoch under sub-setting one

accuracy. For the simple setup, GNM significantly outperforms compared models by increasing the baseline prediction accuracy by 3.1% - 7.4%. On the other hand, GNM is less sensitive to the sample selection and has smaller variance compared to the method by [17]. For the complicated setup, mis-specifying the 'Non-Ignorable' missingness as 'Ignorable' still has big biases even though achieving some improvement against SM. The mean prediction accuracy by GNM is between 3.7% to 4.8% higher than that by GIM.

In both sub-settings, our method always leads to the smallest estimation variance, which is less affected by the selection of labelled nodes. For both setups, higher $\lambda$ value leads to bigger sampling bias, and subsequently there is more significant improvement in the prediction accuracy. Figures 3 and 4 are the boxplots of prediction accuracy obtained from each method under the two model setups. It may intuitively demonstrates the necessity of taking into account missing mechanism in order to achieve higher prediction accuracy on the unlabelled nodes.

We also empirically analyze the computational efficiency of our algorithm. The number of epochs for GNM to achieve convergence in the 50-run real-data experiments is 3 (21), 4 (19), 5 (7), and 6 (3) in Setting 1. Figure 5 summaries the number of iterations for the 2-layer GCN in SM and those for Step 2 of our algorithm at each epoch. It is demonstrated that the computational cost of our GNM model at each epoch is comparable to that of the baseline GCN model.

## Footnotes

[1]Our implementation of GNM can be found at: `https://github.com/BIG-S2/keras-gnm`

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
