[Supplementary Material]

# Supplementary Material for "Graph-Based Semi-Supervised Learning with Nonignorable Nonresponses"

**Fan Zhou[1], Tengfei Li[2], Haibo Zhou[2], Jieping Ye[3], Hongtu Zhu[3]**
Shanghai University of Finance and Economics[1], `zhoufan@mail.shufe.edu.cn`
University of North Carolina at Chapel Hill[2], `tengfei_li@med.unc.edu,zhou@bios.unc.edu`
AI Labs, Didi Chuxing [3], `{yejieping,zhuhongtu}@didiglobal.com`

## 1 Algorithm

---

**Algorithm 1** Gradient Descent-based Joint Estimation Procedure

---

**Input:** $\mathbf{x} \in R^{N \times p}$; $r_i y_i$ for $\forall i$; $A \in R^{N \times N}$

1: **Initialize** $\pi_i^{(0)}, \theta_r^{(0)}, \theta_y^{(0,0)}$; $e = 0$
2: **while** $\sum_{r_i=0} |y_i^{(e)} - y_i^{(e-1)}| / \sum_i 1(r_i = 0) > \varepsilon$ **do**
3:      $e \leftarrow e + 1; w_0, w_1 = 0; \mathscr{L}_1(\theta_y^{best}|\theta_r^{(e-1)}), \widetilde{\mathscr{L}}_2(\theta_r^{best}|\theta_y^{(e)}) = \infty$
4:      **for** $i \leftarrow 0$ to $(M_0 - 1)$ **do**
5:          $\theta_y^{(e,i+1)} \leftarrow \theta_y^{(e,i)} - \gamma_0 \nabla_{\theta_y} \mathscr{L}_1(\theta_y|\theta_r^{(e-1)})$
6:          **if** $\mathscr{L}_1(\theta_y^{(e,i+1)}|\theta_r^{(e-1)}) < \mathscr{L}_1(\theta_y^{best}|\theta_r^{(e-1)})$ **then**
7:              $\theta_y^{best} \leftarrow \theta_y^{(e,i+1)}$
8:          **else**
9:              $w_0 \leftarrow w_0 + 1$
10:             **if** $w_0 > P_0$ **then**
11:                **break**
12:             **end if**
13:          **end if**
14:      **end for**
15:      $\theta_y^e \leftarrow \theta_y^{(e,i)}$
16:      **for** $j \leftarrow 0$ to $(M_1 - 1)$ **do**
17:          $\theta_r^{(e,j+1)} \leftarrow \theta_r^{(e,j)} - \gamma_1 \nabla_{\theta_r} \widetilde{\mathscr{L}}_2(\theta_r|\theta_y^{(e)})$
18:          **if** $\widetilde{\mathscr{L}}_2(\theta_r^{(e,j+1)}|\theta_y^{(e)}) < \widetilde{\mathscr{L}}_2(\theta_r^{best}|\theta_y^{(e)})$ **then**
19:              $\theta_r^{best} \leftarrow \theta_r^{(e,j+1)}$
20:          **else**
21:              $w_1 \leftarrow w_1 + 1$
22:             **if** $w_1 > P_1$ **then**
23:                **break**
24:             **end if**
25:          **end if**
26:      **end for**
27:      $\theta_r^e \leftarrow \theta_r^{(e,j)}$
28: **end while**

---

## 2 Theorem Proofs

### 2.1 Lemma and proof

The proof of Theorem 3.1 is based on the following lemma. Let $supp(\cdot)$ be the support of a domain space.

**Lemma 2.1.** *Under models (2), (3), or (2), (5) (main text), suppose that there exists an instrumental variable $u_i$ in each $x_i = (z_i^T, u_i^T)^T$ such that $f(y_i|\mathscr{G}^A(\boldsymbol{x})_i)$ depends on $u_i$, whereas $P(r_i = 1|y_i, h(x_i))$ does not depend on $u_i$. We let $\boldsymbol{x} = [\boldsymbol{z}, \boldsymbol{u}]$. Our GNM model is identifiable on the PEQ space under the following sufficient Conditions (C1)-(C3):*

*(C1) there exists a set $S \subset supp(Y, \boldsymbol{z})$, such that $P(r_i = 1|y_i, h(z_i); \theta_r) \neq 0$ for each $i$ and all $(Y, \boldsymbol{z}) \in S$ and $\theta_r \in \mathcal{D}(\theta_r)$.*

*(C2) Denote $\theta_{r1} = (\alpha_{r1}, \gamma_1, \phi_1, \theta_{h1})^T$ and $\theta_{r2} = (\alpha_{r2}, \gamma_2, \phi_2, \theta_{h2})^T$. $P(r_i = 1|y_i, h(z_i); \theta_{r1}) = P(r_i = 1|y_i, h(z_i); \theta_{r2})$ for each $i$ and all $(Y, \boldsymbol{z}) \in S \iff \gamma_1^T h(z_i; \theta_{h1}) = \gamma_2^T h(z_i; \theta_{h2})$ holds for all $\boldsymbol{z}$ and each $z_i$.*

*(C3) Denote $\theta_{y1} = (\alpha_1, \beta_1, \theta_{g1})^T$ and $\theta_{y2} = (\alpha_2, \beta_2, \theta_{g2})^T$. We let $\boldsymbol{x}_1 = [\boldsymbol{z}, \boldsymbol{u}_1]$ and $\boldsymbol{x}_2 = [\boldsymbol{z}, \boldsymbol{u}_2]$. If $f(y_i|\mathscr{G}^A(\boldsymbol{x}_1)_i; \theta_{y1})f(y_i|\mathscr{G}^A(\boldsymbol{x}_2)_i; \theta_{y2}) = f(y_i|\mathscr{G}^A(\boldsymbol{x}_1)_i; \theta_{y2})f(y_i|\mathscr{G}^A(\boldsymbol{x}_2)_i; \theta_{y1})$ holds for each $i$ and all $(\boldsymbol{u}_1, \boldsymbol{u}_2)$ and $(Y, \boldsymbol{z}) \in S$, then $\mathscr{G}^A(\boldsymbol{x}; \theta_{g1})\beta_1 = \mathscr{G}^A(\boldsymbol{x}; \theta_{g2})\beta_2$ holds.*

**Proof:** Suppose that the following two equations hold for all $(Y, \mathbf{z}) \in S$ and $(\mathbf{u}_1, \mathbf{u}_2): \mathbf{u}_1 \neq \mathbf{u}_2$, then for each $i$ we have

$$P(r_i = 1|y_i, h(z_i); \theta_{r1})f(y_i|\mathscr{G}^A(\mathbf{x}_1)_i; \theta_{y1}) = P(r_i = 1|y_i, h(z_i); \theta_{r2})f(y_i|\mathscr{G}^A(\mathbf{x}_1)_i; \theta_{y2})$$

$$P(r_i = 1|y_i, h(z_i); \theta_{r2})f(y_i|\mathscr{G}^A(\mathbf{x}_2)_i; \theta_{y2}) = P(r_i = 1|y_i, h(z_i); \theta_{r1})f(y_i|\mathscr{G}^A(\mathbf{x}_2)_i; \theta_{y1}) \quad (1)$$

Multiplying the two equations gives

$$P(r_i = 1|y_i, h(z_i); \theta_{r1})f(y_i|\mathscr{G}^A(\mathbf{x}_1)_i; \theta_{y1})P(r_i = 1|y_i, h(z_i); \theta_{r2})f(y_i|\mathscr{G}^A(\mathbf{x}_2)_i; \theta_{y2})$$

$$= P(r_i = 1|y_i, h(z_i); \theta_{r2})f(y_i|\mathscr{G}^A(\mathbf{x}_1)_i; \theta_{y2})P(r_i = 1|y_i, h(z_i); \theta_{r1})f(y_i|\mathscr{G}^A(\mathbf{x}_2)_i; \theta_{y1})$$

Together with condition (C1), it follows that

$$f(y_i|\mathscr{G}^A(\mathbf{x}_1)_i; \theta_{y1})f(y_i|\mathscr{G}^A(\mathbf{x}_2)_i; \theta_{y2}) = f(y_i|\mathscr{G}^A(\mathbf{x}_1)_i; \theta_{y2})f(y_i|\mathscr{G}^A(\mathbf{x}_2)_i; \theta_{y1})$$

holds for each $i$ and all $(Y, \mathbf{z}) \in S$. Then from condition (C3), we have $\mathscr{G}^A(\mathbf{x}; \theta_{g1})\beta_1 = \mathscr{G}^A(\mathbf{x}; \theta_{g2})\beta_2$ for all $\mathbf{x}$, which implies $f(y_i|\mathscr{G}^A(\mathbf{x}_1)_i; \theta_{y1}) = f(y_i|\mathscr{G}^A(\mathbf{x}_1)_i; \theta_{y2})$ from (3) (main text). Then, we obtain from (1) that

$$P(r_i = 1|y_i, h(z_i); \theta_{r1}) = P(r_i = 1|y_i, h(z_i); \theta_{r2})$$

for each $i$ and all $(Y, \mathbf{z}) \in S$. Together with condition (C2), we have $\gamma_1^T h(x_i; \theta_{h1}) = \gamma_2^T h(x_i; \theta_{h2})$ holds for all $\mathbf{z}$ and each $z_i$. and the identifiability on the PEQ space is obtained.

### 2.2 Proof of Theorem 3.1

**Part (i)**:

Under models (2) and (5) (main text), we prove the identifiability for the binary case when $y \in \{1, -1\}$, while all the derivations can be extended to the more general case. We need to show that for each $i$ and all $(y_i, \mathbf{x}) \in S$,

$$\frac{1}{1 + \exp\{-\alpha_{r1} - \gamma_1^T h(x_i; \theta_{h1}) - \phi_1 y_i\}} \frac{1}{1 + \exp\{-y_i(\alpha_1 + \beta_1^T \mathscr{G}^A(\mathbf{x}; \theta_{g1})_i)\}}$$

$$= \frac{1}{1 + \exp\{-\alpha_{r2} - \gamma_2^T h(x_i; \theta_{h2}) - \phi_2 y_i\}} \frac{1}{1 + \exp\{-y_i(\alpha_2 + \beta_2^T \mathscr{G}^A(\mathbf{x}; \theta_{g2})_i)\}} \quad (2)$$

is equivalent to

$$\alpha_{r1} = \alpha_{r2}, \gamma_1 = \gamma_2, \phi_1 = \phi_2, \alpha_1 = \alpha_2, \beta_1 = \beta_2, \theta_{h1} = \theta_{h2}, \theta_{g1} = \theta_{g2}$$

(2) can be rewritten as

$$e^{-\{\alpha_{r1}+\gamma_1^T h(x_i;\theta_{h1})+\phi_1 y_i\}}+e^{-y_i\{\alpha_1+\beta_1^T\mathscr{G}^A(\mathbf{x};\theta_{g1})_i\}}+e^{-(\alpha_1 y_i+\alpha_{r1})-\phi_1 y_i-\gamma_1^T h(x_i;\theta_{h1})-\beta_1^T\mathscr{G}^A(\mathbf{x};\theta_{g1})_i y_i}$$

$$= e^{-\{\alpha_{r2}+\gamma_2^T h(x_i;\theta_{h2})+\phi_2 y_i\}}+e^{-y_i\{\alpha_2+\beta_2^T\mathscr{G}^A(\mathbf{x};\theta_{g2})_i\}}+e^{-(\alpha_2 y_i+\alpha_{r2})-\phi_2 y_i-\gamma_2^T h(x_i;\theta_{h2})-\beta_2^T\mathscr{G}^A(\mathbf{x};\theta_{g2})_i y_i}$$
(3)

Since (3) holds for all $(y_i, \mathbf{x})$, and from Condition (A1), the only possible solution to (3) is

$$\begin{cases} e^{-\{\alpha_{r1}+\gamma_1^T h(x_i;\theta_{h1})+\phi_1 y_i\}} = e^{-\{\alpha_{r2}+\gamma_2^T h(x_i;\theta_{h2})+\phi_2 y_i\}}, \\ e^{-\{\alpha_1+\beta_1^T\mathscr{G}^A(\mathbf{x};\theta_{g1})_i\}} = e^{-\{\alpha_2+\beta_2^T\mathscr{G}^A(\mathbf{x};\theta_{g2})_i\}}, \\ e^{-(\alpha_1+\alpha_{r1})-\phi_1 y_i-\gamma_1^T h(x_i;\theta_{h1})-\beta_1^T\mathscr{G}^A(\mathbf{x};\theta_{g1})_i} = e^{-(\alpha_2+\alpha_{r2})-\phi_2 y_i-\gamma_2^T h(x_i;\theta_{h2})-\beta_2^T\mathscr{G}^A(\mathbf{x};\theta_{g2})_i} \end{cases}$$

which requires

$$\alpha_{r1} = \alpha_{r2}; \phi_1 = \phi_2; \alpha_1 = \alpha_2; \beta_1^T\mathscr{G}^A(\mathbf{x};\theta_{g1})_i = \beta_2^T\mathscr{G}^A(\mathbf{x};\theta_{g2})_i; \gamma_1^T h(x_i;\theta_{h1}) = \gamma_2^T h(x_i;\theta_{h2}),$$

which concludes the identifiability on the PEQ space.

**Part (ii)**:

Under models (2) and (3) (main text), we prove the identifiability of the parameter when the responses $y$ are continuous. By using Lemma (2.1), Condition (C1) holds due to (2) (main text). Condition (C2) holds due to Condition (A3) in Theorem 3.1. We next give the proof of Condition (C3). We here give the proof of $q = 1$ which can be extended to the general case.

If $f(y_i|\mathscr{G}^A(\mathbf{x}_1)_i;\theta_{y1})f(y_i|\mathscr{G}^A(\mathbf{x}_2)_i;\theta_{y2}) = f(y_i|\mathscr{G}^A(\mathbf{x}_1)_i;\theta_{y2})f(y_i|\mathscr{G}^A(\mathbf{x}_2)_i;\theta_{y1})$ holds for each $i$ and all $(\mathbf{u}_1, \mathbf{u}_2)$ and $(y, \mathbf{z}) \in S$, from (3) (main text), the following equation holds

$$\begin{aligned} &\beta_1^2[(\mathscr{G}^A([\mathbf{z}, \mathbf{u}_1];\theta_{g1}))^2 - (\mathscr{G}^A([\mathbf{z}, \mathbf{u}_2];\theta_{g1}))^2] \\ - \quad &2(y-\alpha_1)\beta_1[\mathscr{G}^A([\mathbf{z}, \mathbf{u}_1];\theta_{g1}) - \mathscr{G}^A([\mathbf{z}, \mathbf{u}_2];\theta_{g1})] \\ = \quad &\beta_2^2[(\mathscr{G}^A([\mathbf{z}, \mathbf{u}_1];\theta_{g2}))^2 - (\mathscr{G}^A([\mathbf{z}, \mathbf{u}_2];\theta_{g2}))^2] \\ - \quad &2(y-\alpha_2)\beta_2[\mathscr{G}^A([\mathbf{z}, \mathbf{u}_1];\theta_{g2}) - \mathscr{G}^A([\mathbf{z}, \mathbf{u}_2];\theta_{g2})] \end{aligned}$$

for all $y$. Together with Condition (A2), we have

$$\beta_1[\mathscr{G}^A([\mathbf{z}, \mathbf{u}_1];\theta_{g1}) - \mathscr{G}^A([\mathbf{z}, \mathbf{u}_2];\theta_{g1})] = \beta_2[\mathscr{G}^A([\mathbf{z}, \mathbf{u}_1];\theta_{g2}) - \mathscr{G}^A([\mathbf{z}, \mathbf{u}_2];\theta_{g2})]$$

and

$$\beta_1[\mathscr{G}^A([\mathbf{z}, \mathbf{u}_1];\theta_{g1}) + \mathscr{G}^A([\mathbf{z}, \mathbf{u}_2];\theta_{g1})] = \beta_2[\mathscr{G}^A([\mathbf{z}, \mathbf{u}_1];\theta_{g2}) + \mathscr{G}^A([\mathbf{z}, \mathbf{u}_2];\theta_{g2})].$$

It follows that

$$\beta_1\mathscr{G}^A([\mathbf{z}, \mathbf{u}_1];\theta_{g1}) = \beta_2\mathscr{G}^A([\mathbf{z}, \mathbf{u}_1];\theta_{g2})$$

and Condition (C3) holds, which concludes the proof.

## 2.3 Proof of Theorem 3.2

To prove the theorem, we use the law of iterated conditional expectation as follows:

$$\begin{aligned} E_{\theta_y}\{\sum_i \frac{r_i}{\pi(y_i, h(x_i))}l(y_i, \mathscr{G}^A(\mathbf{x})_i)\} &= E_{\theta_y}[E\{\sum_i \frac{r_i}{\pi(y_i, h(x_i))}l(y_i, \mathscr{G}^A(\mathbf{x})_i)|Y, \mathbf{x}\}] \\ &= E_{\theta_y}\{\sum_i \frac{E(r_i|Y, \mathbf{x})}{\pi(y_i, h(x_i))}l(y_i, \mathscr{G}^A(\mathbf{x})_i)\} \\ &= E_{\theta_y}\{\frac{\pi(y_i, h(x_i))}{\pi(y_i, h(x_i))}l(y_i, \mathscr{G}^A(\mathbf{x})_i)\} \\ &= E_{\theta_y}\{l(y_i, \mathscr{G}^A(\mathbf{x})_i)\}, \end{aligned}$$
(4)

where (4) holds because

$$E(r_i|Y, \mathbf{x}) = E(r_i|y_i, x_i) = E(r_i|y_i, h(x_i)) = P(r_i = 1|y_i, h(x_i)) = \pi(y_i, h(x_i))$$

# 3 Experiments on 'Citeseer' by comparing GCN and GAT based GNM

We add more experiments on some other dataset, such as 'Citeseer', and explore the finite sample performance of our model by using other state-of-art architectures, such as GAT. Tables 1 and 2 summarize their performance statistics for all model settings on the 'Cora' and 'Citeseer' datasets, respectively. Specifically, 'SM + GCN' and 'SM + GAT' represent the classic 2-layer GCN and GAT models, respectively, whereas 'GNM + GCN' and 'GNM + GAT' are our models by using either GCN or GAT. According to Table 1, we find that 'GNM + GAT' performs slightly better than 'GNM + GCN' for both $\lambda$ settings on 'Cora' dataset. However, the improvement is not significant.

When it comes to the 'Citeseer' dataset, our GNM model can improve the baseline performance by up to 58.5% because of the increased sampling bias (bigger group proportion ratio $\lambda N_0 / N_1$ in the whole population). Different from 'Cora' dataset, 'GNM + GCN' outperforms 'GNM + GAT' in this case and is especially good at handling the more biased setting ($\lambda = 2$). It may indicate that 'GNM + GCN' is less sensitive to the data structure and selection of sampled nodes. Thus, 'GNM + GCN' setting may be preferred when either the missing mechanism or the global distribution of $y$ is unknown.

| $(N_0, N_1)$ | $\lambda$ | Method | Accuracy Mean | SD |
|---|---|---|---|---|
| | 1 | SM + GCN | 0.8683 | 1.98e-2 |
| | | GNM + GCN | 0.8947 | 6.47e-3 |
| | | SM + GAT | 0.8771 | 1.51e-2 |
| (1890, 818) | | GNM + GAT | 0.8968 | 8.65e-3 |
| | 2 | SM + GCN | 0.8052 | 3.26e-2 |
| | | GNM + GCN | 0.8648 | 2.54e-2 |
| | | SM + GAT | 0.8015 | 3.85e-2 |
| | | GNM + GAT | 0.8706 | 1.95e-2 |

Table 1: Prediction Accuracy for 'Cora'

| $(N_0, N_1)$ | $\lambda$ | Method | Accuracy Mean | SD |
|---|---|---|---|---|
| | 1 | SM + GCN | 0.8537 | 2.47e-2 |
| | | GNM + GCN | 0.8981 | 7.95e-3 |
| | | SM + GAT | 0.8076 | 7.98e-2 |
| (2626,701) | | GNM + GAT | 0.8785 | 2.97e-2 |
| | 2 | SM + GCN | 0.5295 | 1.24e-1 |
| | | GNM + GCN | 0.8325 | 7.09e-2 |
| | | SM + GAT | 0.5898 | 1.42e-2 |
| | | GNM + GAT | 0.8090 | 6.45e-2 |

Table 2: Prediction Accuracy for 'Citeseer'