[Reviews · NeurIPS 2019]

Reviewer 1



1. Most of the existing graph based SSL methods assume that the nodes are labeled at random, however authors claim that the probability of missingness may depend on the unobserved data conditioned upon the observed data. I think this claim should be motivated well enough, at least to me it was not entirely clear why this is important. If authors can provide some scenarios which can help understand the claim, it will be beneficial for the readers. 2. Overall, the proposed model is novel and is supported by strong theory. However, the experimental analysis can be improved. One of the baselines the authors consider is "SM", but do not mention the paper in which it is proposed. Authors show results for only one real world dataset i.e., Cora dataset. They should produce results on multiple real world datasets. 3. Going by the motivation of the work, incorporating missing responses is important for graph based SSL. However, authors do not compare the proposed model with state of the art graph based SSL methods like GAT (Velickovic et al., ICML 2018) etc. [Velickovic et al., ICML 2018] Graph Attention Networks 4. Minor points: -- "vertexes" -> "vertices" -- Not sure if using gradient descent qualifies as a contribution. [Comments after reading the author response] I thank the authors for providing additional results. I'm satisfied with the author response and I vote for acceptance of the paper.

Reviewer 2



I think this paper is well prepared. I have the following comments. (1) I think one important contribution is in theoretical aspect. Nevertheless, I cannot judge the proposed results is based on the previous works or these results in developed by the authors themselves. Thus, I suggest the authors to highlight their unique contributions. (2) I am concerned about the compiutational complexity of the proposed algorithm. Thus, I suggest the authors to analyze it. (3) The paper assumed that r_i follows a Bernoulli distribution, how about other distributions?

Reviewer 3



Inference under non-ignorable missing response variables is considered for graph embedding setting. Ignoring these missing variables leads to biased estimation under MNAR, so it is important to consider the missing mechanism explicitly. This paper focuses on missingness of response variables only. So, missingness of graph embedding is not considered. In fact, graph embedding is implemented as the ordinary GCN in the experiments. Therefore, the novelty is basically for handling the missingness of the response variables only, and such method is not quite novel in statistics.

[Author Response · NeurIPS 2019]

**Reviewer #1**: We appreciate many insightful comments from this reviewer.

1. ***Some scenarios which can help understand the claim that nonignorable missing is important in nodes labeling.***

We have included more scenarios in the paper. Here are three of them. (i) A researcher is more likely to label the

documents in a citation network that fall into the categories he is more familiar with. (ii) When predicting the traffic of

a road network, sensors used to collect data are usually set up at the intersections with larger traffic flow. (iii) Medical

studies usually recruit a higher proportion of diseased people than in the whole population to reduce experiment cost.

2. ***Explanation of baseline model 'SM'***

In this paper, SM stands for the standard two-layer GCN model. Our GNM model can be reduced to SM when it only

contains the outcome model $Y|\mathbf{x}$ given in (3) or (5), with the weights in loss (10) being 1 for all samples.

3. ***Experimental on multiple real world datasets and comparison with other SSL methods.***

In the last few days, we have tried very hard to carry out more experiments on other datasets including 'Citeseer', and

explore the performance of our method using other state-of-art architecture such as GAT. Table 1 below compares

  all model settings on 'Citeseer'. The 'SM + GCN' and 'SM + GAT' represent the classic 2-layer GCN and GAT

| $(N_0, N_1)$ | $\lambda$ | Accuracy | | |
| | | Method | Mean | SD |
| --- | --- | --- | --- | --- |
| | 1 | SM + GCN | 0.8537 | 2.47e-2 |
| | | GNM + GCN | 0.8981 | 7.95e-3 |
| | | SM + GAT | 0.8076 | 7.98e-2 |
| (2626,701) | | GNM + GAT | 0.8785 | 2.97e-2 |
| | 2 | SM + GCN | 0.5295 | 1.24e-1 |
| | | GNM + GCN | 0.8325 | 7.09e-2 |
| | | SM + GAT | 0.5898 | 1.42e-2 |
| | | GNM + GAT | 0.8090 | 6.45e-2 |

Table 1: Mean Prediction Accuracy for 'Citeseer'

Figure 1: Boxplot of RMSEs in real data analysis

models, whereas 'GNM + GCN' and 'GNM + GAT' are our models using GCN or GAT architectures to estimate

function $\mathscr{G}^A(\mathbf{x}; \theta_g)$. Because of the increased sample bias (bigger $\lambda N_0/N_1$), our model (GNM + GCN, GNM + GAT)

can improve the baseline performance (SM + GCN, SM + GAT) by up to 58.5%. We also find that GAT does not

significantly outperform GCN, but it is better at handling the extremely biased case. Thus, 'GNM + GAT' setting may

be preferred when either the missing mechanism or the global distribution of $y$ is unknown.

**Reviewer #2**: We appreciate many insightful comments from this reviewer.

1. ***Highlight the unique contributions.***

Compared with the existing literatures, our unique contributions mainly include: (i) being first to explore the label

sampling mechanisms in graph-based semi-supervised learning; (ii) a novel identifiablity for neural network architecture

and easily checked sufficient conditions; and (iii) the integration of gradient descent (GD) algorithm with traditional

estimating equations to estimate parameters for deep learning architectures. Regarding (ii), the existing literature

focuses on identifiability conditions for traditional statistical models, which do not hold in deep learning scenario.

2. ***More analysis about computational issue.***

In each epoch, the complexity (consider one layer GCN only) in Steps 2 and 3 is $O(|E|pq)$ according to Kitf and

Welling (2016), where $|E| < N^2$ is the number of edges; the complexity of Step 4 is $O(Np)$ if $h$ is a one fully

connected layer. We also analyze the computation efficiency of our algorithm. The number of epochs for GNM to

achieve convergence in the 50-run real-data experiments: 3(21), 4(19), 5(7), and 6(3). Figure 1 summaries iteration

times of the 2-layer GCN in SM and the step 2 of our algorithm at each epoch.

3. ***The paper assumed that $r_i$ follows a Bernoulli distribution, how about other distributions?***

We can definitely consider other distributions for $r_i$ and other link functions, such as Probit. We find that model in (2)

can approximate other models with small bias and be robust to model misspecification due to its high flexibility.

**Reviewer #3**: We appreciate many insightful comments from this reviewer.

***Novelty in terms of methodology particularly for the graph embedding setting is to be clearly explained.***

Graph embedding is used to capture the latent patterns of graphs, which help to improve its prediction performance.

Different from traditional statistical methods, we employ an end-to-end architecture to jointly estimate all the parameters

instead of extracting features and fitting the prediction model separately. On the other hand, we use gradient descent

(GD) for parameter estimations to handle the potential deep learning structures in our model other than using traditional

optimization approach such as Newton-Raphson algorithm. Moreover, we incorporate a deep neural network $h(x_i; \theta_h)$

together with the exponential tilting model to more precisely approximate the unknown probability of missing in

practice. We also make some theoretical contributions by defining a novel identifiability in prediction-equivalence

quotient space for neural network architecture. Please also see the first response to Reviewer 2.

[Meta-Review · NeurIPS 2019]

This paper considers graph-based semi-supervised learning when missing labels are not missing at random (NMAR), in other words, the absence of a label is 'nonignorable'. It introduces a graphical neural network that models the relationship between the presence or absence of a label and the labels of its neighbors. The paper also proves that this model is identifiable. The reviewers agree that studying NMAR labels in the context of neural graph embeddings is a novel and important topic. They make several suggestions for improvement, including comparing with recent work such as Velickovic et al., ICML 2018.